# COVID-19 associated changes in HIV service delivery over time in Central Africa: Results from facility surveys during the first and second waves of the pandemic

**Ajeh Rogers** [1]*, **Ellen Brazier**[2], **Anastase Dzudie**[1], **Adebola Adedimeji**[3], **Marcel Yotebieng**[3], **Benjamin Muhoza**[4], **Christella Twizere**[5], **Patricia Lelo**[6], **Dominique Nsonde**[7], **Adolphe Mafoua**[8], **Athanase Munyaneza**[4], **Patrick Gateretse**[5], **Merlin Diafouka**[7], **Gad Murenzi**[4], **Théodore Niyongabo**[5], **Kathryn Anastos**[3], **Denis Nash**[2]

1 Clinical Research Education Networking and Consultancy, Yaoundé, Cameroon, 2 Institute for Implementation Science in Population Health (ISPH), Graduate School of Public Health and Health Policy, City University of New York, New York, NY, United States of America, 3 Albert Einstein College of Medicine, Bronx, NY, United States of America, 4 Einstein-Rwanda Research and Capacity Building Program, Rwanda Military Hospital, Kigali, Rwanda, 5 Centre National de Reference en Matière de VIH/SIDA (CNR), Bujumbura, Burundi, 6 Kalembelembe Pediatric Hospital, Kinshasa, Democratic Republic of Congo, 7 Centre de Traitement Ambulatoire de Brazzaville, Brazzaville, Republic of Congo, 8 Centre de Traitement Ambulatoire de Pointe Noire, Pointe Noire, Republic of Congo

* ajehrogers@gmail.com

**Data Availability Statement:** All relevant data are within the paper and its Supporting Information files.

## Abstract

### Introduction

The COVID-19 pandemic has impacted population health around the globe, directly and indirectly. The objective of this study was to document changes in HIV care associated with the COVID-19 pandemic at selected clinics in Central Africa, along with clinic-level strategies for minimizing disruptions in HIV care and treatment for people with HIV (PWH).

### Methods

A 51-item questionnaire on COVID-19 pandemic-associated changes in HIV service delivery was completed by clinicians involved in HIV care at 21 clinics in five countries participating in Central Africa International epidemiology Databases to Evaluate AIDS (CA-IeDEA). The survey was completed at two timepoints: June-July 2020 and October 2020 to February 2021. Descriptive statistics were used to characterize changes in HIV care and related services.

### Results

While 81% of sites reported at least one negative consequence of COVID-19 for clinic operations during the first survey, none reported suspending antiretroviral therapy (ART) initiation services for new patients, and 24% reported adopting telemedicine. In the follow-up survey, fewer sites (48%) reported at least one disruption to clinic operations, and more sites reported mitigation strategies, including expanding rapid ART initiation services and providing extra supplies of ART medications to reduce visit frequency. In the follow-up survey, more sites, especially in Rwanda, reported stockouts of commodities, including HIV

**Funding:** Research reported in this publication was supported by the National Institutes of Health's National Institute of Allergy and Infectious Diseases (NIAID), the Eunice Kennedy Shriver National Institute of Child Health & Human Development (NICHD), the National Cancer Institute (NCI), the National Institute on Drug Abuse (NIDA), the National Heart, Lung, and Blood Institute (NHLBI), the National Institute on Alcohol Abuse and Alcoholism (NIAAA), the National Institute of Diabetes and Digestive and Kidney Diseases (NIDDK), the Fogarty International Center (FIC), the National Library of Medicine (NLM), and the Office of the Director (OD) under Award Number U01AI096299 (Central Africa-IeDEA). The content is solely the responsibility of the authors and does not necessarily represent the official views of the National Institutes of Health. The funders, however, had no role in study design, data collection and analysis, decision to publish, or preparation of the manuscript.

**Competing interests:** The authors have declared that no competing interests exist.

and viral load testing and HIV pre-exposure prophylaxis. More than one-fifth of sites reported stockouts of second- or third-line ART at each survey timepoint.

## Conclusions

While the initial wave of the COVID-19 pandemic resulted in concerning disruptions to HIV service delivery at CA-IeDEA sites, most of these disruptions attenuated over time, and many sites introduced measures to help PWH avoid frequent visits to the clinic for care and medications. The impact of HIV commodity stockouts and clinic mitigation strategies on treatment outcomes needs to be assessed.

## Introduction

The COVID-19 pandemic has impacted population health around the globe, both directly and indirectly [1]. The first case of COVID-19 in Africa was reported on February 14, 2020, just two weeks after the novel coronavirus, SARS-CoV2, was declared a public health emergency of international concern by the World Health Organization (WHO) [2]. Within three months, COVID-19 had spread to almost all countries in Africa, and the first wave peaked in mid-July 2020 [2].

Early modeling studies raised concerns about potential indirect effects of the pandemic on the availability, accessibility and quality of basic health services, including HIV services [3–6], particularly given the high burden of both infectious and non-communicable diseases in sub-Saharan Africa [7]. A modelling group convened by the WHO and UNAIDS estimated that a six-month disruption of antiretroviral therapy could lead to more than 500,000 extra deaths from AIDS-related illnesses, including tuberculosis, in sub-Saharan Africa in 2020–2021 [8]. In view of the population health impacts of service delivery disruptions for people living with and at risk for HIV, the WHO, UNAIDS, the United States President's Emergency Plan for AIDS Relief (PEPFAR), and other partners called for changes in service delivery to minimize disruptions in HIV care and treatment and to mitigate risks of HIV-related care-seeking [9–12]. Recommended measures included the provision of multi-month supplies of antiretroviral therapy (ART) drugs and preexposure prophylaxis (PrEP), along with addressing geographic and financial barriers to access, introducing telemedicine and virtual consultations to minimize COVID-19-related risks, and postponing non-urgent care visits, along with increased attention to respiratory and surface hygiene in service delivery settings.

While a recent analysis [2] has documented the introduction of public health and social measures (PHSMs) in Africa to mitigate the spread of COVID-19, there are limited published data on the impact of the pandemic on the availability, accessibility and quality of HIV services in the region. The goal of this study was to document the extent to which the COVID-19 pandemic has impacted HIV care and treatment at selected clinics in Central Africa, along with clinic-level mitigation strategies put in place to minimize service delivery disruptions and risks for people with HIV (PWH).

## Methods

### Ethical review

This study protocol was reviewed and designated a non-human subjects operational/quality improvement project by the Vanderbilt University Medical Center (VUMC) Institutional

Review Board. As the survey collected only site-level data and did not involve human subjects, informed consent was not required.

## Study design

During the first year of the COVID-19 pandemic, we conducted cross-sectional surveys on the availability of HIV care and related services at 21 HIV clinics in five Central African countries at two different time points. All 21 HIV care and treatment clinics that participate in the Central Africa cohort of the International epidemiology Databases to Evaluate AIDS (IeDEA) research consortium [13] completed the survey at each time point, including three clinics each in Burundi and Cameroon, one in the Democratic Republic of Congo (DRC), two in the Republic of Congo (ROC) and 12 in Rwanda (100% response rate at each time point).

## Data collection

At each clinic, HIV care providers completed a 51-item questionnaire designed to capture changes in the clinic's environment, operations and HIV-related services that were related to the COVID-19 pandemic. The first round of data collection (Round 1) was conducted in June-July 2020, which coincided with the peak of the first wave of the pandemic in much of Africa [2]. The follow-up data collection (Round 2) was conducted during a four-month period (October 2020 to February 2021) as part of a larger survey administered across all sites participating in the global IeDEA research consortium [14]. Because the follow-up survey coincided with a rapidly-evolving second wave of the pandemic in many African countries [2], a relatively long timeframe was given for survey completion. At both survey time points, the survey was distributed as an online questionnaire, using REDCap electronic data capture tools [15] hosted at Vanderbilt University Medical Center, and using a paper form for sites that did not have reliable internet access. Surveys completed via the paper form were entered into REDCap by Central Africa IeDEA data managers in each country.

## Survey domains

The survey collected information about whether sites' locality had been subject to any government restrictions or public health measures to limit the spread of COVID-19, as well as changes in HIV service delivery that were instituted because of the pandemic. Questions explored changes in facility- and community-based services for HIV testing, ART initiation, ART retention, and routine patient care, as well as stockouts of HIV-related medications and laboratory testing supplies. In addition, information was collected on routine HIV service provision prior to the pandemic, including criteria for and timing of ART initiation, practices related to ART readiness counseling, frequency of ART refills, availability of viral load testing and usual turnaround time for viral load test results. Survey data were linked to existing descriptive information about each clinic participating in Central Africa IeDEA, including information about the age groups and urban vs. rural residence of the patients served, and facility type or level (e.g., health center vs. hospital).

## Statistical analysis

Descriptive statistics were used to characterize COVID-19-related changes in HIV care and service delivery, including the extent to which clinics and clinic localities were subject to COVID-19-related restrictions, and the duration of lockdowns or service delivery suspensions. We examined the proportion of sites reporting COVID-19-related disruptions and changes in

service delivery in each survey. Sites that did not provide a given service prior to the COVID-19 pandemic were excluded from the denominators used to calculate descriptive statistics.

All data cleaning and analyses were performed using SAS 9.4 (SAS Institute, Cary, NC).

## Results

Of the 21 HIV clinics participating in Central Africa IeDEA, twelve (57%) serve patients from both urban and rural areas, two (9.5%) serve a predominantly urban patient population and seven (33%) primarily serve patients residing in rural areas (Table 1). All Central Africa IeDEA sites provide HIV care for pediatric patients, along with adolescents and youth (ages 10–24 years) and adults (ages 20+ years).

All Central Africa IeDEA sites had adopted the World Health Organization's "Treat All" guidance prior to the COVID-19 pandemic, providing ART to all people diagnosed with HIV irrespective of CD4 count or clinical status. Ten (48%) sites reported that they had offered same-day ART initiation (i.e., initiate patients on ART on the day they enroll in HIV care) prior to the pandemic, with eight (38%) reporting that they initiated patients on ART within one week of care enrollment and the remaining sites (3/21) initiating patients on ART 2–4 weeks after enrollment. The majority (58%) of primary-level sites (i.e., health centers) offered same-day ART initiation services prior to the pandemic compared with only one-third of hospitals. Most sites (14 or 67%) reported requiring only one ART readiness counseling session prior to ART initiation, and the typical frequency of ART refills for stable patients was a three-month supply (18 or 86%). Less than half of the sites (8 or 38%) had onsite viral load testing prior to the COVID-19 pandemic, and the turnaround time for viral load test results ranged from 0–7 days (29%), 7 to 14 days (19%), to more than 2 weeks (52%).

Changes in clinic operations attributed to the COVID-19 pandemic during each round of the survey are shown in Table 2, with data shown by country in S1 and S2 Tables. During Round 1 of data collection, almost half of the Central Africa IeDEA sites (10 or 48%)—primarily in DRC, ROC and Rwanda—reported that their locality had been subject to COVID-19-related restrictions on travel, service provision or business operations. By Round 2, 15 sites (71%), including sites from all five countries, reported that their locality had ever been subject to such restrictions, with the duration of lockdowns and restrictions ranging from less than one month (5 sites), 2–3 months (6 sites), five or more months or ongoing (2 sites), or not recalled (2 sites).

During both survey rounds, few sites reported interruptions in recording data related to patient care or the withdrawal or suspension of support from non-governmental organizations (NGOs) involved in facility-based HIV care at their sites. However, most sites reported that the pandemic had negatively impacted care, with 81% of sites reporting at least one negative consequence of COVID-19 for clinic operations during Round 1, including eight sites (38%, including 5/12 in Rwanda, 1/1 in DRC and 2/2 in Republic of Congo) reporting reduced hours or days for HIV services, and almost half (9 sites or 43%, including sites in all five countries), reporting reduced availability of HIV care providers because of re-assignment of staff to support the COVID-19 response (three sites) or reduced staff availability because of COVID-19-related illness, quarantine or isolation (eight sites). In contrast, during Round 2, fewer sites (10 or 48%) reported disruptions to clinic operations, with only one site reporting that hours or days for HIV service delivery were reduced, and five sites (24%, 1/3 in Cameroon and 2/12 in Rwanda), reporting reduced provider availability.

The number of sites reporting that space at their hospital/clinic was reconfigured to accommodate COVID-19-related services decreased slightly between the two surveys, from eight to seven, with sites in Burundi, Cameroon, and Rwanda reporting space reconfigurations during

**Table 1. Characteristics of Central Africa IeDEA sites.**

| | All | Health Center | Regional, provincial or university hospital |
|---|---|---|---|
| | N (%) | n (%) | n (%) |
| **Site characteristic** | **21** | **12 (57%)** | **9 (43%)** |
| Country | | | |
| Burundi | 3 (14%) | 1 (8%) | 2 (22%) |
| Cameroon | 3 (14%) | (0%) | 3 (33%) |
| Democratic Republic of Congo (DRC) | 1 (5%) | (0%) | 1 (11%) |
| Republic of Congo | 2 (10%) | (0%) | 2 (22%) |
| Rwanda | 12 (57%) | 11 (92%) | 1 (11%) |
| Population served (residence) | | | |
| Rural | 7 (33%) | 4 (33%) | 3 (33%) |
| Urban | 2 (10%) | 2 (17%) | (0%) |
| Mixed | 12 (57%) | 6 (50%) | 6 (67%) |
| Population served (age groups) | | | |
| Adults (20+ years) | 21 (100%) | 12 (100%) | 9 (100%) |
| Adolescents/youth (10–24 years) | 21 (100%) | 12 (100%) | 9 (100%) |
| Pediatric patients (<10 years) | 21 (100%) | 12 (100%) | 9 (100%) |
| ART initiation criteria in 2019 | | | |
| Treat All | 21 (100%) | 12 (100%) | 9 (100%) |
| Other | (0%) | (0%) | (0%) |
| Timing of ART initiation in 2019 | | | |
| Same day | 10 (48%) | 7 (58%) | 3 (33%) |
| 1–7 days | 8 (38%) | 5 (42%) | 3 (33%) |
| 18–14 days | 3 (14%) | (0%) | 3 (33%) |
| ART counseling sessions required in 2019 | | | |
| 0 | 2 (10%) | 2 (17%) | (0%) |
| 1 | 14 (67%) | 8 (67%) | 6 (67%) |
| 2 | 3 (14%) | 2 (17%) | 1 (11%) |
| 3 | 2 (10%) | (0%) | 2 (22%) |
| ART refill frequency for stable patients in 2019 | | | |
| Every 1–2 months | 1 (5%) | (0%) | 1 (11%) |
| Every 3 months | 18 (86%) | 11 (92%) | 7 (78%) |
| Every 6 months | 2 (10%) | 1 (8%) | 1 (11%) |
| Viral load testing in 2019 | | | |
| Point of care/Same day testing | 11 (52%) | 7 (58%) | 4 (44%) |
| Turnaround time for viral load test results in 2019 | | | |
| 0–7 days | 6 (29%) | 5 (42%) | 1 (11%) |
| 14 days | 4 (19%) | 3 (25%) | 1 (11%) |
| 15–30 days | 9 (43%) | 4 (33%) | 5 (56%) |
| 30–60 days | 2 (10%) | (0%) | 2 (22%) |

ART: Antiretroviral therapy; IeDEA: International epidemiology Databases to Evaluate AIDS

both survey points. At Round 2, decreases were observed in the number of sites reporting increased use of personal protective equipment, from 91% to 76% (primarily in Burundi, ROC and Rwanda), and in the number reporting increased use of telemedicine (24% to 19%).

Changes in HIV service delivery attributed to the COVID-19 pandemic are shown in Table 3 (with data shown by country in S3 and S4 Tables). No sites reported suspending or postponing the enrollment of new patients in HIV care at either survey time point. However,

**Table 2. COVID-19 pandemic-related changes in clinic environment or operations at Central Africa IeDEA sites, 2020–2021.**

| Change in clinic environment or operations | Round 1 (June-July,2020) N = 21 | Round 2 (Oct 2020-Feb 2021) N = 21 |
|---|---|---|
| Geographic area surrounding this HIV clinic subject to any form of COVID-19 restrictions on travel, service provision, or business operations | 10 (48%) | 15 (71%) |
| Duration of lockdowns/restrictions (as of 2nd round survey) | | |
| ≤1 month | - | 5 (24%) |
| 2–3 months | - | 6 (29%) |
| 5+ months or ongoing | - | 2 (10%) |
| Do not know/recall | - | 2 (10%) |
| Not applicable | - | 6 (29%) |
| At least one negative impact on clinic operations | 17 (81%) | 10 (48%) |
| Decreases in the number of hours or days of service delivery for HIV patients | 8 (38%) | 1 (5%) |
| Reduced staffing among HIV care providers | 9 (43%) | 5 (24%) |
| Re-assignment of providers to assist with the COVID-19 response | 3 (14%) | 3 (14%) |
| Reduced provider availability because COVID-19-related illness, self-isolation, or quarantine | 8 (38%) | 3 (14%) |
| Reconfiguration of hospital/clinic space to accommodate COVID-19-related services | 8 (38%) | 7 (33%) |
| Interruptions or changes in recording of data (either paper or electronic records) related to clinical management of patients | 1 (5%) | 2 (10%) |
| Withdrawal/suspension of activities of non-governmental partners that support care provision in the clinic (N = 20)* | 3 (15%) | 1 (5%) |
| Increased use of personal protective equipment (masks, gloves, gowns, etc.) by HIV clinic staff | 19 (91%) | 16 (76%) |
| Increased use of telemedicine (i.e., consultations by phone/web) in HIV-related care | 5 (24%) | 4 (19%) |

IeDEA: International epidemiology Databases to Evaluate AIDS

Round 1: June—July 2020; Round 2: October 2020—February 2021.

* Sites where the service was not available prior to the COVID-19 pandemic excluded from denominator.

in Round 1, half of the sites (11/21), including sites from all countries except Cameroon, reported that HIV testing/diagnostic services had been suspended or reduced, with the majority of these sites (8/11) being located in Rwanda. In contrast, in Round 2, no sites reported interruptions to HIV testing services. In addition, although a few sites (4/21, all in Rwanda) initially reported that they had suspended or postponed non-urgent appointments for HIV patients, none of the sites reported that they were currently postponing non-urgent appointments for HIV patients during Round 2.

None of the sites reported shutting down the ART clinic or suspending ART initiation services for newly enrolling patients at either survey time point. In addition, at each survey time point, most sites (62% and 71%, respectively) reported giving patients extra ART refills to reduce the need to return to the clinic. There were small increases in the number of sites also reported expanding same-day and rapid ART initiation, from three sites (in Burundi and Rwanda) during Round 1 to five sites in Round 2 (in Burundi, Cameroon and Rwanda). However, no site reported establishing community-based ART pickup locations for their patients at either survey time point, and, while one-third of the sites in Rwanda initially reported that they had streamlined ART adherence counseling, only two sites reported streamlined counseling during Round 2.

**Table 3. Changes in HIV-related services and capacity at Central Africa IeDEA sites, 2020–2021.**

| HIV Service | Round 1 (June-July,2020) N = 21 | Round 2 (Oct 2020—Feb 2021) N = 21 |
|---|---|---|
| **HIV testing and care enrollment** | | |
| Suspension or decreases in HIV testing/diagnostic services | 11 (52.4%) | (0%) |
| Suspension or postponement of the enrollment of new patients in HIV care | (0%) | (0%) |
| Suspension or postponement of non-urgent appointments for HIV patients | 4 (19.0%) | (0%) |
| **ART services** | | |
| ART clinics have been suspended or shut down | (0%) | (0%) |
| ART initiation services have been suspended | (0%) | (0%) |
| Patients given extra supplies/refills of ART to reduce the frequency of refills | 13 (61.9%) | 15 (71.4%) |
| ART pick-up points designated in the community | (0%) | (0%) |
| Expansion of same-day/rapid ART initiation | 3 (14.3%) | 5 (23.8%) |
| Adherence counseling streamlined | 4 (19.0%) | 2 (9.5%) |
| **Viral load testing services** | | |
| Sample collection suspended | 7 (33.3%) | 1 (4.8%) |
| VL samples no longer accepted | (0%) | 1 (4.8%) |
| Longer turn-around time | 3 (14.3%) | 5 (23.8%) |
| Other (staffing shortages, lack of transport for samples) | 2 (9.5%) | (0%) |
| **Community-based services** | | |
| Suspension of activities of NGO partners that support community-based programs for patients enrolled in HIV care at the clinic | 6 (30%) | 0 (0%) |
| Withdrawal/suspension of activities of non-governmental partners that support care provision in the clinic | 6 (30%) | 0 (0%) |
| Community-based HIV testing suspended (N = 17)* | 11 (64.7%) | 3 (17.6%) |
| Community-based ART refills suspended (N = 18)* | 4 (22.2%) | 1 (5.6%) |
| Community-based support group meetings/activities suspended (N = 20)* | 15 (75.0%) | 4 (20.0%) |
| Community-based tracing of patients lost to follow-up (LTFU) suspended | 15 (71.4%) | 1 (4.8%) |
| **Stockouts** | | |
| PrEP | (0%) | 1 (5.0%) |
| HIV test kits | 1 (4.8%) | 2 (9.5%) |
| First-line ART | (0%) | (0%) |
| Second-line ART | 2 (9.5%) | 1 (4.8%) |
| Third-line ART (N = 19)* | 3 (15.8%) | 3 (15.8%) |
| Supplies for viral load testing | 3 (14.3%) | 5 (23.8%) |

ART: Antiretroviral therapy; IeDEA: International epidemiology Databases to Evaluate AIDS; PrEP: Pre-exposure prophylaxis

Round 1: June—July 2020; Round 2: October 2020—February 2021.

* Sites where the service was not available prior to the COVID-19 pandemic excluded from denominator.

In Round 1, 7/12 of the Rwandan sites reported that the pandemic had resulted in the suspension of blood sample collection for viral load testing services, and three sites (1 DRC and 2 in Rwanda) reported longer turnaround times for obtaining viral load test results. In Round 2, almost one-quarter (24%) of the sites (in Burundi and Rwanda) reported longer than normal turnaround times for viral load testing, and two sites in Rwanda also reported that viral load

testing was suspended or that HIV viral load samples were no longer accepted by the virology laboratory.

More than half of the sites, in all five countries, reported that community-based HIV services had been affected by the pandemic, particularly during Round 1, with 65%, 75% and 71%, respectively, reporting the suspension of community-based HIV testing services, support group meetings and activities, and community tracing of patients who missed care visits. While the proportions reporting the suspension of these community-based services was markedly lower in Round 2 (18%, 20% and 5%, respectively), some gaps in community-based programs persisted at sites in Rwanda.

A minority of sites reported stockouts of HIV-related commodities (e.g., pre-exposure prophylaxis or PrEP; HIV test kits; first-line, second-line, or third-line ART medications; and viral load testing supplies) at each survey round. However, in Round 2, slightly more sites reported stockouts of PrEP (1 site vs. 0), HIV test kits (2 vs 1), and viral load testing supplies (5 vs 3), compared with Round 1. Stockouts of viral load testing supplies were only reported by sites in Rwanda.

## Discussion

Through repeat surveys on COVID-19-related changes to HIV service delivery across 21 clinics in five countries in Central Africa, we found that the initial response to COVID-19 pandemic negatively impacted clinic operations and HIV service delivery at a substantial proportion of sites. Disruptions included decreased hours of service delivery, reduced provider availability and the suspension of certain HIV services and programs. Disruptions in care during the initial phase of the pandemic appeared to be most acute for HIV testing services at the clinic and community levels and viral load testing, as well as for community-based programs, such as community support groups and patient tracing activities. These interruptions in service delivery could pose significant barriers for a subset of individuals (e.g., undiagnosed persons testing for HIV and unsuppressed individuals who may require enhanced adherence support and/or a change in treatment regimen).

A minority of sites reported that non-urgent appointments for HIV patients were suspended or postponed during Round 1, however, no site reported suspending or postponing the enrollment of new patients in HIV care or the suspension of either routine ART clinics or ART initiation services for new patients at either time point. Similarly, while 11 sites initially reported a decrease or suspension of HIV testing and diagnostic services, no site reported such interruptions during Round 2. Also encouraging, many sites reported adopting various strategies to minimize COVID-19 risks for patients and limit disruptions in routine care, including streamlined ART adherence counselling, expansion of same-day or rapid ART initiation, the provision of extra supplies of ART medications to reduce the frequency of refills. Of concern, however, the use personal protective equipment appeared to have decreased in the Round 2 survey—findings that align with other research showing that public health and social measures aimed at mitigating the spread of COVID-19 were markedly lower during the second wave of the pandemic in Africa compared with the first [2].

While it was encouraging that a few sites reported increased use of telemedicine, its use decreased between the two surveys, These findings are consistent with research highlighting barriers to the adoption and use of telemedicine in sub–Saharan Africa and other low-resource settings, which range from socioeconomic, digital literacy and linguistic barriers among patients to infrastructure, technology, and regulatory obstacles for health systems [16, 17]— barriers that should assessed and addressed as part of initiatives to improve pandemic readiness in these settings [18].

While some in the global HIV community recommended decentralized delivery of ART outside of health facilities to minimize the necessity of attending clinics early in the pandemic [19], none of the sites in CA-IeDEA reported establishing community-based ART pick-up points at either time point. These results may reflect the fact that most of the HIV clinics participating in CA-IeDEA are located in urban settings, where travel distances and costs associated with HIV care-seeking are generally lower, compared with rural areas [20–22]. Of concern, however, is that many sites reported that crucial activities, such as community tracing of patients who missed HIV care visits, had been suspended, particularly during the first survey, conducted early in the pandemic.

The apparent expansion of certain mitigation strategies (e.g., the provision of extra ART supplies and the expansion of same-day or rapid ART initiation services) was encouraging. In addition, some of the initial disruptions to HIV service delivery appeared to have eased, with fewer sites reporting reduced hours or days of service delivery, or disruption or suspension of both clinic- and community-based services, including patient tracing activities—changes that may reflect increasing adaptation to the pandemic over time [18], as well as the easing of government restrictions on travel and other operations, particularly in Rwanda, DRC and ROC, where the initial response to the pandemic included more stringent COVID-19 containment and health measures, compared with Burundi and Cameroon [23]. Other research has shown that more than 75% of patients who are lost to follow-up can be reengaged in ART through patient tracing; thus, the resumption of these activities may be especially critical for reengaging patients who were became lost to follow-up during the initial wave of the pandemic [24, 25].

The follow-up survey results suggest longer-term effects of the pandemic on HIV-related supply chains, as more sites reported stockouts of essential commodities and more reported longer turn-around times for obtaining viral load testing results. There was also a small decrease in reported use of PPE, which could reflect either shortages of these items, or decreased attention to PPE use—either of which could increase risks of COVID-19 infection among both patients and providers. These health system challenges could negatively impact HIV care outcomes if not resolved, as shortages of antiretroviral drugs have been associated with poor ART adherence, the development of drug resistance, and increases in HIV related death [26–29]. These consequences could be worsened by ineffective viral load monitoring systems, especially in resource constrained settings in sub-Saharan Africa [29].

Our findings accord with other recent research in sub-Saharan Africa, including an analysis of service delivery data in South Africa, which showed that while ART services were generally maintained during the initial phase of the pandemic, HIV testing and other laboratory services were more heavily impacted [12, 30]. Our findings are also consistent with research showing that while stringent measures were enacted during the first wave of the COVID-19 pandemic in Africa, fewer countries adopted such measures during the second wave, despite the increased severity of the second wave and the serious public health implications for individuals and health systems [2].

Covering two time points at 21 HIV clinics from five countries, our longitudinal study gives a broad perspective of the measures implemented at HIV care and treatment clinics in response to the COVID-19 pandemic over time. As continuous assessment of the direct and indirect impact of COVID-19 on other priority public health programs can inform mitigation efforts, our study provides timely data for the ongoing COVID-19 response.

The generalizability of our study may be limited by the non-representativeness of the sites that participate in the Central Africa IeDEA research consortium. While our findings may be reflective of the pandemic response strategies adopted by national and local governments in each country, which may have direct or indirect implications for HIV care and services, our estimates may not be generalizable to all HIV clinics in the study countries. A further

limitation is that our data were self-reported by staff at participating sites during both rounds of the survey, and different staff may have responded to the two surveys at a given site; there may be recall errors related to the COVID-19 response measures that were adopted at different time points, and different providers may have different perceptions of the measures put in place to mitigate the pandemic. Also, this study did not assess changes in patient attendance in the clinic or in funding allocations for HIV care, which could also impact service delivery and HIV treatment outcomes.

The results of this study are encouraging in terms of the level and duration of disruptions to HIV service delivery. However, our findings underscore the need to closely monitor and mitigate the short- and long-term impact of COVID-19 pandemic on health programs in order to preserve health gains achieved over decades, a concern strongly emphasized by the WHO [31]. Further research is needed to understand how care-seeking among PWH has been affected, particularly as public health measures put in place to limit the spread of COVID-19, in tandem with patients' concern about their risk of infection, may have limited access to and utilization of available services [32]. Accordingly, it is critical to assess the impact of the COVID-19 pandemic on programmatic outcomes, including new HIV infections, retention of patients in HIV care and treatment, and viral suppression among patients on ART. As patient-level data from CA-IeDEA sites become available it will also be important to understand how patient outcomes are associated with the mitigation efforts undertaken at these clinics and to identify best practices for supporting care retention and sustained viral suppression in the face of the ongoing and evolving COVID-19 pandemic.

## Conclusions

Data from this survey indicate that while the COVID-19 pandemic negatively impacted service delivery at HIV care and treatment clinics in Central Africa, the potential for care disruptions attenuated after the initial phase of the pandemic. In addition, some sites have introduced strategies to mitigate risks to both HIV patients and providers, including measures to reduce non-urgent visits to the clinic and the provision of additional ART supplies. Further research is needed to assess the impact of COVID-19 on key programmatic outcomes, including retention in care, ART adherence and viral load suppression among PWH.

## Supporting information

**S1 Table. Changes in clinic environment or operations at Central Africa IeDEA sites, by country, Round 1 (June—July 2020).** ART: Antiretroviral therapy; IeDEA: International epidemiology Databases to Evaluate AIDS. ** Sites where the service was not available prior to the COVID-19 pandemic excluded from denominator.
(DOCX)

**S2 Table. Changes in clinic environment or operations at Central Africa IeDEA sites, by country, Round 2 (October 2020—February 2021).** ART: Antiretroviral therapy; IeDEA: International epidemiology Databases to Evaluate AIDS; PrEP: Pre-exposure prophylaxis. ** Sites where the service was not available prior to the COVID-19 pandemic excluded from denominator.
(DOCX)

**S3 Table. Effects of COVID-19 on HIV-related services and capacity at Central Africa IeDEA sites, by country, Round 1 (June—July 2020).** ART: Antiretroviral therapy; IeDEA: International epidemiology Databases to Evaluate AIDS; PrEP: Pre-exposure prophylaxis. ** Sites where the service was not available prior to the COVID-19 pandemic excluded from

denominator.
(DOCX)

**S4 Table. Effects of COVID-19 on HIV-related services and capacity at Central Africa IeDEA sites, by country, Round 2 (Oct 2020—Feb 2021).** ART: Antiretroviral therapy; IeDEA: International epidemiology Databases to Evaluate AIDS; PrEP: Pre-exposure prophylaxis. ** Sites where the service was not available prior to the COVID-19 pandemic excluded from denominator.
(DOCX)

**S1 File.**
(DOCX)

## Acknowledgments

We formally thank all Central Africa IeDEA sites for their participation in the survey. A complete list of acknowledgments can be found in the S1 File.

## Author Contributions

**Conceptualization:** Ajeh Rogers, Ellen Brazier, Anastase Dzudie.

**Data curation:** Ellen Brazier.

**Formal analysis:** Ajeh Rogers, Ellen Brazier.

**Methodology:** Ajeh Rogers, Ellen Brazier, Anastase Dzudie, Adebola Adedimeji.

**Project administration:** Ajeh Rogers, Ellen Brazier.

**Validation:** Ajeh Rogers.

**Writing – original draft:** Ajeh Rogers, Ellen Brazier.

**Writing – review & editing:** Ajeh Rogers, Ellen Brazier, Anastase Dzudie, Adebola Adedimeji, Marcel Yotebieng, Benjamin Muhoza, Christella Twizere, Patricia Lelo, Dominique Nsonde, Adolphe Mafoua, Athanase Munyaneza, Patrick Gateretse, Merlin Diafouka, Gad Murenzi, Théodore Niyongabo, Kathryn Anastos, Denis Nash.

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
