## [Decision Letter · Decision Letter 0]

25 Apr 2022

PONE-D-22-02487COVID-19 associated changes in HIV service delivery over time in Central Africa: Results from facility surveys during the first and second waves of the pandemicPLOS ONE

Dear Dr. Rogers,

Thank you for submitting your manuscript to PLOS ONE. After careful consideration, we feel that it has merit but does not fully meet PLOS ONE’s publication criteria as it currently stands. Therefore, we invite you to submit a revised version of the manuscript that addresses the points raised during the review process.

ACADEMIC EDITOR:

Your paper was of interest to the reviewers. As you can see below, the reviewers identified both strengths and weaknesses in your paper. The reviewers noted important areas of your paper that require careful attention. From my own reading of your paper, I am in agreement with the reviewers that your paper will make an important contribution and that your paper will benefit from a thoughtful revision. Please revise your paper in accordance with the reviews particularly looking at grammar and punctuation. In addition, please include a discussion on lessons learned from inter and intra country variations. 

Please be sure to include a cover letter with your revision that addresses all of the reviews and comments as well as my comments, point-by-point. Your revision should also carefully follow the journal style as presented in the Instructions for Authors available at the journal website.  

We look forward to receiving your revised manuscript.

Kind regards,

Brian C. Zanoni, MD

Academic Editor

PLOS ONE

Journal Requirements:

“Research reported in this publication was supported by the National Institutes of Health’s National Institute of Allergy and Infectious Diseases (NIAID), the Eunice Kennedy Shriver National Institute of Child Health & Human Development (NICHD), the National Cancer Institute (NCI), the National Institute on Drug Abuse (NIDA), the National Heart, Lung, and Blood Institute (NHLBI), the National Institute on Alcohol Abuse and Alcoholism (NIAAA), the National Institute of Diabetes and Digestive and Kidney Diseases (NIDDK), the Fogarty International Center (FIC), the National Library of Medicine (NLM), and the Office of the Director (OD) under Award Number U01AI096299 (Central Africa-IeDEA). The content is solely the responsibility of the authors and does not necessarily represent the official views of the National Institutes of Health.”

“Research reported in this publication was supported by the National Institutes of Health’s National Institute of Allergy and Infectious Diseases (NIAID), the Eunice Kennedy Shriver National Institute of Child Health & Human Development (NICHD), the National Cancer Institute (NCI), the National Institute on Drug Abuse (NIDA), the National Heart, Lung, and Blood Institute (NHLBI), the National Institute on Alcohol Abuse and Alcoholism (NIAAA), the National Institute of Diabetes and Digestive and Kidney Diseases (NIDDK), the Fogarty International Center (FIC), the National Library of Medicine (NLM), and the Office of the Director (OD) under Award Number U01AI096299 (Central Africa-IeDEA). The content is solely the responsibility of the authors and does not necessarily represent the official views of the National Institutes of Health”

Additional Editor Comments (if provided):

Your paper was of interest to the reviewers. As you can see below, the reviewers identified both strengths and weaknesses in your paper. The reviewers noted important areas of your paper that require careful attention. From my own reading of your paper, I am in agreement with the reviewers that your paper will make an important contribution and that your paper will benefit from a thoughtful revision. Please revise your paper in accordance with the reviews. In particular please review for grammar and punctuation errors and include lessons learned from intra and inter country variations as also mentioned by reviewer 1.

Please be sure to include a cover letter with your revision that addresses all of the reviews and comments as well as my comments, point-by-point. Your revision should also carefully follow the journal style as presented in the Instructions for Authors available at the journal website.

Reviewers' comments:

Reviewer's Responses to Questions

**Comments to the Author**

1. Is the manuscript technically sound, and do the data support the conclusions?

Reviewer #1: Partly

Reviewer #2: Yes

2. Has the statistical analysis been performed appropriately and rigorously? 

Reviewer #1: Yes

Reviewer #2: Yes

3. Have the authors made all data underlying the findings in their manuscript fully available?

Reviewer #1: Yes

Reviewer #2: Yes

4. Is the manuscript presented in an intelligible fashion and written in standard English?

Reviewer #1: Yes

Reviewer #2: Yes

5. Review Comments to the Author

Reviewer #1: The paper sought to provide changes effected in HIV service delivery over time in selected countries of Central Africa, in the wake of the COVID-19 pandemic. It presents and discusses results from facility surveys during the first and second waves of the pandemic in 2020 and 2021, along with clinic-level mitigation strategies for minimizing disruptions to HIV care and treatment, in some instances. Following a critical review and appraisal of the paper, the authors should address the following, with a view to strengthen the message and utility of the findings:

1. Key results in the abstract and the text allude to negative outcomes related to HIV service delivery in light of the pandemic, such as stockouts of HIV testing commodities and antiretroviral treatments (ART), in particular during the second, follow up survey. However much less is discussed in terms of strategies adopted for mitigation of these negative outcomes, if any. It ably dwells on consequences, and less on changes effected, or adaptation outcomes. This is important for establishing what worked and drawing lessons for the future in these and/ or in other countries similarly affected.

2. While disruptions due to the pandemic, as noted, are important, more attempt should be made to not only document, but also explain any differences observed, between and within the countries studied. Lessons can also be drawn from any inter and intra country variations in responses adopted to mitigate consequences of the COVID-19 pandemic. This thus takes full advantage of the multi-country approach.

Reviewer #2: This is an interesting and valuable contribution to the literature. I have some minor suggestions to help make the piece even stronger:

1. Did all of the site characteristics remain the same pre and during the pandemic?

2. You mentioned that the survey captured, "...issuance and timing of any government restrictions and public health measures to limit the spread of COVID-19 in the locality of each clinic". Can you add any more of this information to the manuscript?

3. In Table 3, I wouldn't label the title "Effects" given you aren't actually showing a causal relationship here in your table. Please identify other words that don't have the same connotation.

4. In your tables, try and provide more space in your rows so the data can more easily be read; perhaps shrink the font.

6. PLOS authors have the option to publish the peer review history of their article (what does this mean?). If published, this will include your full peer review and any attached files.

Reviewer #1: **Yes: **Prof. Michael T. Mbizvo

Reviewer #2: No

---

## [Author Response · Author response to Decision Letter 0]

5 Jul 2022

We thank the Editor for considering our manuscript. In our revised manuscript, we have aligned the formatting of the manuscript and file names with PLOS ONE’s style templates.

We appreciate the Editor’s feedback and advice. We have clarified that the study was determined to be non-human subjects research for which informed consent was not required (lines 165-172).

“Research reported in this publication was supported by the National Institutes of Health’s National Institute of Allergy and Infectious Diseases (NIAID), the Eunice Kennedy Shriver National Institute of Child Health & Human Development (NICHD), the National Cancer Institute (NCI), the National Institute on Drug Abuse (NIDA), the National Heart, Lung, and Blood Institute (NHLBI), the National Institute on Alcohol Abuse and Alcoholism (NIAAA), the National Institute of Diabetes and Digestive and Kidney Diseases (NIDDK), the Fogarty International Center (FIC), the National Library of Medicine (NLM), and the Office of the Director (OD) under Award Number U01AI096299 (Central Africa-IeDEA). The content is solely the responsibility of the authors and does not necessarily represent the official views of the National Institutes of Health.”

Thank you for clarifying these points. We have removed the funding information from the Acknowledgements section. For readability we have also shortened the Acknowledgements and presented the full list of sites and individuals involved as Supplementary Information.

“Research reported in this publication was supported by the National Institutes of Health’s National Institute of Allergy and Infectious Diseases (NIAID), the Eunice Kennedy Shriver National Institute of Child Health & Human Development (NICHD), the National Cancer Institute (NCI), the National Institute on Drug Abuse (NIDA), the National Heart, Lung, and Blood Institute (NHLBI), the National Institute on Alcohol Abuse and Alcoholism (NIAAA), the National Institute of Diabetes and Digestive and Kidney Diseases (NIDDK), the Fogarty International Center (FIC), the National Library of Medicine (NLM), and the Office of the Director (OD) under Award Number U01AI096299 (Central Africa-IeDEA). The content is solely the responsibility of the authors and does not necessarily represent the official views of the National Institutes of Health”

We have removed all funding-related text from the manuscript and confirm that no changes are needed to the funding statement.

As noted above, we have added the full name of the IRB which determined that the study was non-human subjects research for which informed consent was not required (lines 165-172).

Additional Editor Comments (if provided):

Your paper was of interest to the reviewers. As you can see below, the reviewers identified both strengths and weaknesses in your paper. The reviewers noted important areas of your paper that require careful attention. From my own reading of your paper, I am in agreement with the reviewers that your paper will make an important contribution and that your paper will benefit from a thoughtful revision. Please revise your paper in accordance with the reviews. In particular please review for grammar and punctuation errors and include lessons learned from intra and inter country variations as also mentioned by reviewer 1.

Please be sure to include a cover letter with your revision that addresses all of the reviews and comments as well as my comments, point-by-point. Your revision should also carefully follow the journal style as presented in the Instructions for Authors available at the journal website.

Dear Editor, 

Thank you for your consideration and feedback on our paper. We have worked to address each of the reviewers’ suggestions in our revision, and have addressed the reviewers’ feedback point by point, below. All line numbers refer to the clean version of the manuscript. We have also made additional edits, in response to your suggestions to align the manuscript with the journal requirements, while also editing the manuscript to improve the readability of the manuscript and address grammar and punctuation errors.

Sincerely, 

Rogers Ajeh

Responses to reviewers’ comments

Reviewer 1

1. Key results in the abstract and the text allude to negative outcomes related to HIV service delivery in light of the pandemic, such as stockouts of HIV testing commodities and antiretroviral treatments (ART), in particular during the second, follow up survey. However much less is discussed in terms of strategies adopted for mitigation of these negative outcomes, if any. It ably dwells on consequences, and less on changes effected, or adaptation outcomes. This is important for establishing what worked and drawing lessons for the future in these and/ or in other countries similarly affected.

Thank you very much for this comment. We had noted in our abstract that more sites reported expanding rapid ART initiation services and providing extra supplies of ART medications, however, we had not characterized these changes as mitigation strategies. In our revision, we have explicitly described these practices as mitigation strategies (lines 33-35). We have also added a statement about the proportion of sites reporting increased use of telemedicine during the first survey (line 32). 

2. While disruptions due to the pandemic, as noted, are important, more attempt should be made to not only document, but also explain any differences observed, between and within the countries studied. Lessons can also be drawn from any inter and intra country variations in responses adopted to mitigate consequences of the COVID-19 pandemic. This thus takes full advantage of the multi-country approach.

Thank you for this comment. As noted among our study limitations, the clinics participating in IeDEA may not be representative of HIV clinics in each country, and findings may reflect recall errors, particularly as there were numerous and frequent changes in national policy responses during the first year of the COVID-19 pandemic.1 Because of our small sample size and these limitations, we are hesitant to over-interpret some of the observed variations within and between countries reflected in our study. That said, we have given more attention to some of the differences observed between countries in presenting the results, and we have highlighted them further in the discussion—noting in particular that some changes may reflect the easing of government restrictions in settings, such as Rwanda, ROC, and DRC, which had more stringent containment measures early in the pandemic than Burundi and Cameroon (lines 269-272). 

Reviewer 2

1. Did all of the site characteristics remain the same pre and during the pandemic?

We appreciate the reviewer’s consideration of our manuscript and feedback. While we sought to assess pandemic-related impacts on HIV service delivery, we did not attempt to document changes in site characteristics, such as health facility level, urban vs. rural location, population served, and other characteristics reported in Table 1. As we note in our Methods section (lines 107-110), survey data were linked with existing descriptive information about participating sites—data that are stored in a separate database. Although IeDEA does update this database on participating sites every two to three years, site-level characteristics, such as facility designation, urbanicity, or patient population rarely change in these settings, and the interval of time between the two study surveys was very short. For characteristics related to ART initiation and viral load testing practices, we explored practices in 2019, prior to the start of the COVID-19 pandemic, as a reference point for understanding how practices changed during the first year of the pandemic. 

2. You mentioned that the survey captured, "...issuance and timing of any government restrictions and public health measures to limit the spread of COVID-19 in the locality of each clinic". Can you add any more of this information to the manuscript?

Thank you for raising this point. As described in Table 2 and lines 147-148, at each survey timepoint, we explored whether sites’ locality or area of coverage had been subject to restrictions on travel, service provision or business operations because of the pandemic, and we sought to estimate the duration of these restrictions. We did not attempt to document different types of restrictions or public health measures because we were concerned about recall errors during a period of frequent changes in public health policy and pandemic mitigation measures. To avoid the perception that we aimed to document specific types of government restrictions or public health measures, we have modified the wording in our Methods section (lines 100-101), explaining that the survey gathered information on whether sites’ localities “had been subject to any government restrictions or public health measures to limit the spread of COVID-19.” 

3. In Table 3, I wouldn't label the title "Effects" given you aren't actually showing a causal relationship here in your table. Please identify other words that don't have the same connotation.

Thank you for this suggestion. We have modified the table title, changing it to: Changes in HIV-related services and capacity at Central Africa IeDEA sites during the first- and second round surveys, 2020-2021. 

4. In your tables, try and provide more space in your rows so the data can more easily be read; perhaps shrink the font.

Thank you for this suggestion. We have reduced the font size for each table and increased the spacing. Additionally, to improve readability and to be consistent with similar site survey research2 published in PLoS One, we have removed the decimal points in reporting percentages throughout both the manuscript and all tables.

References

1. Coronavirus Pandemic (COVID-19). OurWorldInData.org; 2020. https://ourworldindata.org/coronavirus. Accessed June 27, 2022.

2. Parcesepe AM, Lancaster K, Edelman EJ, et al. Substance use service availability in HIV treatment programs: Data from the global IeDEA consortium, 2014-2015 and 2017. PLoS ONE. 2020;15(8):e0237772.

---

## [Decision Letter · Decision Letter 1]

19 Sep 2022

COVID-19 associated changes in HIV service delivery over time in Central Africa: Results from facility surveys during the first and second waves of the pandemic

PONE-D-22-02487R1

Dear Dr. Rogers,

We’re pleased to inform you that your manuscript has been judged scientifically suitable for publication and will be formally accepted for publication once it meets all outstanding technical requirements.

Kind regards,

Brian C. Zanoni, MD

Academic Editor

PLOS ONE

Additional Editor Comments (optional):

The authors have appropriate responded to the reviewer comments and the manuscript is acceptable for publication.

Reviewers' comments:

Reviewer's Responses to Questions

**Comments to the Author**

1. If the authors have adequately addressed your comments raised in a previous round of review and you feel that this manuscript is now acceptable for publication, you may indicate that here to bypass the “Comments to the Author” section, enter your conflict of interest statement in the “Confidential to Editor” section, and submit your "Accept" recommendation.

Reviewer #1: All comments have been addressed

2. Is the manuscript technically sound, and do the data support the conclusions?

Reviewer #1: Yes

3. Has the statistical analysis been performed appropriately and rigorously? 

Reviewer #1: Yes

4. Have the authors made all data underlying the findings in their manuscript fully available?

Reviewer #1: No

5. Is the manuscript presented in an intelligible fashion and written in standard English?

Reviewer #1: Yes

6. Review Comments to the Author

Reviewer #1: The authors have responded and addressed my comments.

7. PLOS authors have the option to publish the peer review history of their article (what does this mean?). If published, this will include your full peer review and any attached files.

Reviewer #1: **Yes: **Professor Michael Mbizvo

---

## [Editor Report · Acceptance letter]

25 Oct 2022

PONE-D-22-02487R1 

COVID-19 associated changes in HIV service delivery over time in Central Africa: Results from facility surveys during the first and second waves of the pandemic 

Dear Dr. Rogers:

I'm pleased to inform you that your manuscript has been deemed suitable for publication in PLOS ONE. Congratulations! Your manuscript is now with our production department. 

Kind regards, 

on behalf of

Dr. Brian C. Zanoni 

Academic Editor

PLOS ONE